# Experimental Study on Early Strength and Hydration Heat of Spodumene Tailings Cemented Backfill Materials

**DOI:** 10.3390/ma15248846

**Published:** 2022-12-11

**Authors:** Shunchun Deng, Lang Liu, Pan Yang, Caixin Zhang, Yin Lv, Lei Xie

**Affiliations:** 1School of Energy Engineering, Xi’an University of Science and Technology, Xi’an 710054, China; 2Research Center for Mine Functional Filling Technology, Xi’an 710054, China

**Keywords:** spodumene tailings, cemented tailing backfill, hydration heat, X-ray diffraction, scanning electron microscope

## Abstract

Spodumene tailing is the associated solid waste of extracting lithium from spodumene. With the increase in the global demand for lithium resources, its emissions increase yearly, which will become a key factor restricting the economic development of the mining area. Mechanical and hydration reactions, as well as the microstructure of early CSTB, are studied under different tailings–cement ratios (TCR) and solid mass concentration (SC) conditions. The results show that the uniaxial compressive strength of early CSTB has a negative exponential correlation with the decrease in TCR and a positive correlation with the increase in SC: when the age of CSTB increases to 7 days, the strength increases with the rise in SC in an exponential function, and the sensitivity of strength to TCR is higher than that of SC. Compared to other tailings cemented backfill materials, the addition of spodumene tailings reduces the sulfate ion concentration and leads to a new exothermic peak (i.e., the third exothermic peak) for the hydration exotherm of CSTB. Additionally, with the increase in TCR or decrease in SC, the height of the third exothermic peak decreases and the occurrence time is advanced. At the same time, the duration of induction phase was prolonged, the period of acceleration phase was shortened, and the total amount of heat released was significantly increased. The decrease in TCR or the increase in SC led to the rise in the number of hydration products which can effectively fill the internal pores of CSTB, enhance its structural compactness, and increase its compressive strength. The above study reveals the influence of TCR and SC on the early strength, hydration characteristics, and microstructure of CSTB and provides an essential reference for the mix design of underground backfill spodumene tailings.

## 1. Introduction

Lithium is known as “white oil” for its excellent physical and chemical properties. Lithium plays an irreplaceable role in battery energy, aerospace industries, medicine, and health [1,2]. Global lithium demand is growing steadily at an annual rate of 8–11%, and the total global market is expected to reach 1 million tons in 2025 [3,4]. One of the most critical extraction sources of lithium is spodumene. However, the content of lithium in natural spodumene is relatively low, and a large amount of spodumene tailings are generated during the extraction process [5,6]. China is a major lithium producer, accounting for about 10% of the global lithium production capacity, with nearly 2 million tons of spodumene tailings produced in 2020. Currently, the treatment of spodumene tailings is mainly handled in situ, and the continuous discharge of a large number of tailings causes environmental safety hazards to the air, soil, and water in the surrounding areas and affects human health [7,8,9]. With the rise of the new energy industry, the demand for lithium will continue to increase, and more spodumene tailings will be produced in the future. Effective disposal of spodumene tailings has become an urgent problem to be solved.

Scholars have undertaken a great deal of research to promote the resource utilization of spodumene tailings. This research mainly focuses on ceramic materials, geopolymers, cement mortar aggregate, etc. Yang et al. [10] found that the chemical composition of spodumene tailings contains albite and microcline. The chemical composition is consistent with green ceramic materials and can be utilized to prepare porous ceramics. Using spodumene flotation powder can absorb methylene blue and provide good flexural and compressive strength, apparent porosity, and water absorption. Patrick et al. [11] prepared incombustible lightweight porous ceramics from spodumene tailings and glass wool waste. Such ceramics can be a satisfactory substitute for lightweight building materials for high-rise buildings. However, the preparation of ceramics requires that the tailing size be kept within a specific range to reduce the grinding cost. The uneven distribution of spodumene tailing size may limit the industrial application of large-scale ceramic preparation [12,13]. The polymer composites prepared by Patrick et al. [14], which use a large number of spodumene tailings, have good thermal stability. The form of tailings added (such as grinding, fractionation, or original) will change the material’s thermal shrinkage, compactness, and strength. Yang et al. [12] prepared a green geopolymer with high bending strength using alkali-activated spodumene flotation tailings and metakaolin. Additionally, this treatment method provides a novel idea for resource utilization of spodumene tailings. However, the cost of this geological polymer is relatively high, which is not conducive to its promotion. In addition, Wu et al. [15] used spodumene flotation tailings as the aggregate of cement mortar. They found that increasing tailings’ particle size can effectively improve artillery’s mechanical properties, providing a reference for using spodumene as the aggregate of cement mortar.

As an effective means of solid waste management, tailings cemented backfill has attracted many scholars to study different tailings of cemented backfill materials [16,17]. Fridjonsson et al. [18] sought a more thorough and complete understanding of the function of cemented tailings backfill material as a support structure in mining operations. The changes in pore size and distribution during the hydration of backfill materials are also studied. Behera et al. [19] studied the physical and chemical properties of the mixture of lead–zinc grinding tailings and fly ash and proved the feasibility of using them for underground backfill. Qiu et al. [20] used microorganisms to induce carbonate precipitation as a new auxiliary cementation technology for iron tailings, and the backfill strength of iron tailings treated by this method was significantly improved. However, the existing research focuses on using spodumene tailings in ceramic production or geopolymer materials, such methods is too expensive to achieve the purpose of large-scale use of the waste of tailings. There are no other studies using spodumene tailings as filling materials. If tailings are used for filling, they can meet the strength requirements of conventional metal mines, not only not needing to increase the cost of tailings treatment but also achieving the purpose of protecting the environment. The hydration reaction of the backfill material, especially the early hydration reaction, directly affects the strength development and the later durability performance. The early hydration reaction mechanism and the development law of the hydration reaction products are the focus of early hydration process research [21]. At the same time, the development of thr early strength of filling materials has important practical significance for on-site production, affecting the mining production cycle. The hydration reaction and microstructure evolution of the filling material is the internal driving force for strength formation. The difference in physical and chemical properties of different material components directly affects the hydration process of backfill materials [22,23]. Accurately understanding the change law of the hydration process and the evolution law of microstructure of cemented backfill materials is conducive to analyzing the development law of early strength, thus controlling the backfill material strength formation process and providing reasonable process parameters [24,25,26,27]. However, the early strength development, exothermic hydration rule, and microstructure formation mechanism of CSTB still need to be studied.

This paper explores the influence of different tailings–cement ratios (TCR) and solid mass concentration (SC) on the early strength of CSTB. Additionally, it analyzes its hydration characteristics through the isothermal calorimetry test and studies its microstructure through X-ray diffraction and scanning electron microscope tests to further reveal the development mechanism of strength. It has the further objectives of promoting the utilization of spodumene tailings for mine backfill and providing a theoretical basis for the proportioning design of CSTB materials.

## 2. Materials and Methods

### 2.1. Experimental Material

#### 2.1.1. Tailings

Spodumene tailing was obtained from the general sand outlet of a spodumene mine and used to prepare CSTB samples. The particle size of tailings was measured by a Malvern Mastersizer 2000 laser particle size analyzer. Figure 1a shows the particle size distribution of tailings; the tailing D_10_ is 43.34 μm, D_60_ is 167.29 μm, D_90_ is 305.94 μm, and the nonuniformity coefficient Cu = 7.06. It is generally believed that when Cu ≥ 5, the tailings have good compactness and grading. However, if Cu is too large, it may indicate that the middle particle size is missing, and so the curvature coefficient Cc is also used for evaluation. The curvature coefficient 1 < Cc = 1.28 < 3 indicates that the tailings are well graded and are beneficial to the bonding of tailings particles in the filling process [28]. The chemical composition of tailings (Table 1) shows that the main chemical components of tailings are SiO_2_, Al_2_O_3_, and Na_2_O, that the content is more than 90%, and that higher oxide content may have a good gelling effect [29]. Figure 1b shows the X-ray diffraction (XRD) pattern. The tailings contain albite (Na_2_O Al_2_O_3_ 6SiO_2_, PDF#10-0393), auartz (SiO_2_, PDF#46-1045) and muscovite (KAl_2_(AlSi_3_O_10_) (OH)_2_, PDF#07-0025). They are mainly composed of aluminates or silicates of silicon, aluminum, and sodium and sodium oxides. Above all, spodumene tailing is a comparatively ideal cemented filling material. The analysis results are consistent with the results of chemical element analysis.

#### 2.1.2. Binder and Water

The binder used is ordinary Portland cement P·O 42.5 grade ordinary Portland cement, and the chemical composition analysis results are shown in Table 1. It can be seen from Table 1 that the main chemical components are CaO, SiO_2_, Al_2_O_3,_ and MgO, and that the total amount is 92.12%.

The water used for CSTB is ordinary tap water.

### 2.2. Sample Preparation

Depending on field practice and reading of the literature [30,31], the tailing cement ratio of 4:1, 6:1, 8:1, 10:1, and 12:1 were selected as the research object of this paper. Used solid mass concentrations (SC) of 68%, 70%, 72%, 74%, and 76% were selected. Table 2 shows the evaluated compositions detailing the cement, tailings, and water amount. Mix the tailings, cement, and water according to a certain proportion and pour it into the UJZ-15 planetary cement mortar mixer for stirring. Then, pour the stirred CSTB into the standard cylindrical (φ50 mm×h100 mm) abrasive tool and demold after 48 h. All test pieces, after demolding, were placed in a traditional curing environment with a curing temperature of 20 ± 2 °C and relative humidity of 95% ± 1% for curing.

### 2.3. Methods

Through mechanical experiments, isothermal calorimetry, XRD, and scanning electron microscopy (SEM) experiments, the compressive strength, hydration process, and mineralogical composition of CSTB are analyzed.

According to the standard testing method of mechanical properties on ordinary concrete (GB/T 50081-2002), the CSTB specimen with a certain curing age was tested by WDW (20-1000) electronic universal testing machine. The loading rate is 1 mm/min, and the final strength of CSTB is the average of the strength of the three samples. According to ASTM C 1702 [32], put 15 g dry sample (containing a certain proportion of tailings and cement) powder into a glass ampoule, add water and stir for 2 min, and seal with a plastic plug. Then, put in the TAM air isothermal calorimeter (Thermometric AB, New Castle, DE, USA,) to measure the normalized heat flow (N) and cumulative heat (Q) of CSTB. The test temperature is set at 20 °C, and the test time is 72 h. SEM observed the microstructure and morphology of CSTB. A small part was taken from the inner center of the cured specimen to be broken and immersed in ethanol to prevent hydration. The material was then quickly dried in a drying oven at 60 °C. The dried samples were taken into small pieces, and the samples were sprayed with gold and then tested by a TESCAN MIRA4 field emission environment scanning electron microscope. XRD can determine the type of hydration products. Take a small number of dried samples and grind them in an agate mortar. After grinding until there is no particle feeling, use the Bruker AXS D8 X-ray diffractometer produced by Bruker company in Germany to determine the type of hydration products of CSTB by XRD. The tube current and voltage of XRD equipped with a Cu-Kα X-ray source were 40 mA and 40 kV, respectively. The diffraction angle (2 θ) is 5–90°, and the scanning speed is 6°/min.

## 3. Results and Analysis

### 3.1. Compressive Strength Analysis

#### 3.1.1. Effect of TCR on Compressive Strength of CSTB

Figure 2 shows the relationship between the early strength of CSTB and TCR. It can be observed from Figure 2 that the compressive strength of CSTB is negatively correlated with TCR when the SC is constant. With the continuous decrease in TCR, the intensity of CSTB gradually increased, and this change did not change with age. Taking SC = 76% as an example, when TCR = 12, 10, 8, 6, and 4, the strength is 0.163 MPa, 0.330 MPa, 0.529 MPa, 0.749 MPa, and 1.832 MPa, respectively, at the curing age of 3 days, and the power is increased by 0.167 MPa, 0.199 MPa, 0.22 MPa, and 1.083 MPa, respectively. This is because, with the increase in TCR, the lower the cement content in CSTB is and the smaller the hydration reaction will be. Therefore, the fewer hydration products that will be generated through the hydration reaction, the weaker the connection between tailings particles is, and the more pores there are, resulting in a CSTB intensity reduction [33].

It is worth noting that the lower the TCR is, the more pronounced the strength increase will be under the same SC. The test results are fitted exponentially according to the change law of CSTB strength with TCR. Except for the linear correlation coefficient *R*^2^ of the strength fitting curve with SC of 68% at 7 days being too low, the other correlation coefficients *R*^2^ are all close to 1. This indicates a robust exponential correlation between TCR and CSTB compressive strength under a constant SC, which can be expressed by Equation (1). It is consistent with the conclusions drawn by other scholars [34,35]:(1)$$y=a+be^{-x/c}$$
where *y* is the compressive strength, MPa; *a*, *b* and *c* are constants depending on TCR and SC factors, and *x* is TCR.

#### 3.1.2. Effect of SC on Compressive Strength of CSTB

Figure 3 shows the relationship between the early strength of CSTB and SC. It can be observed in Figure 3 that, when TCR is constant, the uniaxial compressive strength of CSTB increases with the increase in SC, and the growth rate gradually increases with the rise in SC. When TCR = 4, with SC increased from 68% to 70%, the CSTB’s 3-day strength increased from 0.738 to 0.767 MPa, with an increased rate of 3.9%. Additionally, 7-day strength increased from 0.750 to 0.987 MPa, with an increased rate of 31.6%. When SC increases from 74% to 76%, the 3-day strength increases from 1.200 to 1.832 MPa with a growth rate of 52.7%, and the 7-day strength increases from 2.222 to 3.216 MPa with a growth rate of 44.7%. This can be attributed to the fact that when TCR is constant, SC can affect the cement segregation in CSTB. The reason for the segregation is that there is a difference in the settling speed of large and small particles in water. With the increase in SC, the smaller the cement segregation, the more uniform the particle distribution, and the lower the cement loss. Therefore, the relatively low SC backfill material has a higher content of high SC cement, and the strength of CSTB is relatively high. At the same time, with the decrease in SC, the more significant the difference in the sedimentation velocity of large and small particles, the more discrete the distribution, and the lower the intensity of low SC [36].

Under the same TCR, the CSTB intensities of different SCs were fitted. When the curing age was 3 days, except for the fitting curve of TCR = 4, the appropriate effect of the strengths of different SCs under the other TCRs was not noticeable. This is because, with the increase in TCR, the 3-day strength decreases with the decrease in SC. However, when TCR increases to a certain extent, the influence of SC on the 3-day power decreases, and the intensity change is not apparent enough, resulting in a poor fitting effect [37]. With the increase in curing time, the influence of SC on the strength of CSTB gradually increases. When the age reaches 7 days, the fitting of strength shows the exponential correlation shown in Equation (2):(2)$$\;y=a+be^{\left(\left(x-c\right)/d\right)}$$
where *y* is the compressive strength, MPa; *a*, *b* and *c* are constants depending on TCR and SC factors, and x is SC.

#### 3.1.3. Coupling Effect of TCR and SC on Compressive Strength of CSTB

According to Abrams’ law, the water–cement ratio (W/C) is considered the most critical factor affecting concrete strength. The relationship between compressive strength and W/C is not affected by sand type [38,39,40]. TCR and SC determine the W/C, so the early power of CSTB is closely related to TCR and SC, and its strength increases with the decrease in TCR and the increase in SC, and the lower the TCR, the greater the influence of SC on CSTB. Under a specific SC, the intensity of CSTB changes exponentially with the decrease in TCR. The lower the TCR, the more pronounced the intensity increase; at the same time, under a certain TCR, the CSTB intensity also has an exponential correlation with the rise in SC.

From the above analysis, it can be seen that the strength of CSTB is related to TCR and SC, that the two have different effects on the strength, and that the sensitivity of the strength to TCR and SC is different. In Figure 2, it can be observed that, when the SC is constant, the TCR decreases from 12 to 4 in turn, and the strength increases significantly at each stage of reduction. In Figure 3, at 3 days, except for the strength of CSTB increasing obviously with SC when TCR = 4, the strength of CSTB under other TCR does not increase obviously with SC and even cannot reach the fitting effect. However, with the increase in curing age, the strength of CSTB increases obviously with SC compared with that at 3 days, but it is still not as significant as the effect of TCR on the strength. It can be considered that the sensitivity of CSTB compressive strength is TCR > SC. This is consistent with the research results of some scholars [37,41,42], while the conclusions of other scholars are the opposite [30,43]. This difference can be attributed to the following two points. Firstly, the SC concentration gradient set in this paper is slight, and the CSTB intensity does not change significantly with SC. Secondly, this is related to the particle size distribution of backfill materials and chemical composition. For example, the research of Fu et al. [30] on the tailings of iron ore (setting SC as 65%, 68%, 70%, 73%, 75%, and TCR as 4, 5, 6, 8, 10) shows that the CSTB strength of the whole tailings of this iron ore is more sensitive to SC than TCR. However, the research of Yang et al. [42] on another iron ore (setting SC as 65%, 68%, 70%, 73%, and TCR as 4, 6, 8, 10) has reached the opposite conclusion.

### 3.2. Hydration Exothermic Analysis

Cement hydration is an exothermic process. A series of physical and chemical changes occur after CSTB is mixed with water, and a large amount of heat is released. The amount of heat released is related to the cement’s hydration process [44,45].

The study shows that the hydration stage of CPB can be divided into five stages: (1) dissolution stage, (2) induction stage, (3) acceleration stage, (4) deceleration stage, and (5) slow reaction stage [46,47,48]. The dissolution stage starts from the contact between the backfill material and water. At this time, the hydration reaction is rapid, and a large amount of heat is released. Due to metastable barrier [49] or slow dissolution [50], after the initial fast reaction, the hydration reaction slows down, and the exothermic hydration decrease enters the induction stage. After the induction stage, the hydration rate is accelerated and enters the acceleration stage. The exothermic heat in the acceleration stage is closely related to the nucleation of the generated hydrated calcium silicate gel (C-S-H) [51,52,53]. With the progress of the hydration reaction, the hydration products increase, the small redactable particles, available space and redactable water decrease, and the hydration enter the deceleration stage. After that, due to the consumption of cementitious materials, hydration enters the slow reaction stage.

It is worth noting that, compared with the exothermic hydration curve of ordinary cement-based materials, the exothermic hydration curve of CSTB has a significant feature that a new hydration exothermic peak appears on its hydration exothermic curve (Figure 4). The following two reasons may cause this new exothermic peak. First, the hydration reaction of CSTB can be divided into two parts. One part is the hydration reaction of tricalcium silicate (C3S) contained in cement (the second exothermic peak of ordinary cement-based materials). The other part is the hydration reaction of tailings, the third exothermic peak, which is similar to the exothermic hydration curve produced by adding blast furnace slag [54] or iron tailings [22] to cement. Second, the new exothermic peak is related to the hydration reaction of C3A. In this process, sulfate ion concentration plays a key role. C-S-H initially adsorbs the sulfate ion. When solid sulfate is consumed, the slope of thermal evolution increases and C3A hydration leads to a new exothermic peak [55,56,57,58].

#### 3.2.1. Effect of TCR on the Heat Exotherm of CSTB Hydration

Figure 5 is the hydration reaction exothermic rate curve of different TCRs under the same SC and the total exothermic amount curve of SC = 72%. Table 2 indicates the specific changes in the hydration heat evolution of different TCRs under SC = 72%.

Table 3 demonstrates that CSTB with low TCR entered the induction phase somewhat later, but that the duration of this phase was shorter. For example, when TCR = 12, the induction phase was entered 1.1 h after the beginning of the test, and the duration of this phase was 6.6 h. When TCR was reduced to 4, the induction period was delayed by 0.3 h, but the time was shortened by 1.8 h. In this case, the exothermic hydration curve in Figure 5 c shows that the exothermic angle of low TCR is sharper at this stage. This situation can be explained as follows: when the dissolution reaction proceeds to a certain extent, the clinker particles are wrapped by the hydration film, and the formation of the hydration film hinders the hydration reaction, thus reducing the hydration reaction rate of CSTB, and the hydration reaction enters the induction period. With time, the wrapped clinker particles continue to dissolve and participate in the response slowly, the concentration of hydration products C-S-H or calcium hydroxide (CH) increases, and the pressure difference between the inside and outside of the hydrated film increases. The hydrated film breaks when the concentration rises to a critical point, and the hydration reaction enters an accelerated phase. The material with high TCR contains less cement, the water–cement ratio increases, and the increase in available water reduces the concentration of Ca^2+^ in the pore solution. This leads to a longer time being taken for Ca^2+^ to reach saturation and a longer duration of the induction period [46,60].

In the acceleration stage, it can be observed that the hydration heat release rate curve of high TCR, seen in Figure 5a–e, is gentler. The slower the hydration heat release growth is, the lower the height of the exothermic peak is, and the second exothermic peak completely disappears as TCR rises to 12. Table 3 shows that the second and third exothermic peaks of high TCR are advanced, and the duration of this phase is reduced. The hydration rate in the accelerated degree is related to the heterogeneous nucleation and growth of C-S-H on mineral surfaces, which has been supported by many scholars [61,62,63]. Under the same SC, the cement content of CSTB with high TCR is reduced, and the amount of C-S-H generated is relatively reduced. The hydration heat release rate increases more slowly as the hydration heat release curve is flatter and the hydration heat release rate is lower. Hence, the second heat release peak-to-peak value of high TCR is lower [60].

In the acceleration stage, the second hydration exothermic peak of high TCR appears earlier. The acceleration stage’s duration is shortened, which is mainly related to the crystal nucleation and crystal growth of C-S-H and CH. With the increase in TCR, tailing particles increase and can be used as the nucleii for the precipitation and development of initial hydration products such as C-S-H. Therefore, the rise in TCR accelerates the hydration reaction and shortens the duration of the acceleration phase [64]. In addition, the increase in TCR leads to the rise in the water–cement ratio (W/C), the expansion of free water content per unit cement particle, and the decrease in cement that can participate in the hydration reaction, all of which will accelerate the hydration reaction of cement. Therefore, the second exothermic peak appears in advance [24,26]. This is because the tailing content increases with the increase in TCR, resulting in fewer sulfate ions being dissolved in water and the concentration of sulfate ions in the solution decreases. The time point of sulfate depletion is advanced, and C3A is re-dissolved and hydrated, and so the occurrence time of the third exothermic peak is advanced. In addition, with the advance in the third exothermic peak, the second exothermic peak overlapped with the third exothermic peak, which was more evident at high TCR. When TCR = 12, the second exothermic peak could not be observed. This is the same as what was found by Han et al. [27] and Yang [65], who made the observation that the hydration exothermic peak changes with the increase in iron tailing content.

In the deceleration and slow reaction stages, the heat release rate of the backfill material with high TCR decreases more slowly than that of that backfill material with low TCR, and its heat release curve is relatively flat. The hydration rate of backfill materials with high TCR is still lower than that of backfill materials with low TCR. The diffusion process determines the deceleration of the hydration reaction and the rate of the slow reaction stage and is affected by the size of reaction particles, reaction water, and reaction space. The hydration rate of the low-TCR product is relatively higher than that of the high-TCR product because the cement content is higher. There are more reactive particles in the later stage [66].

The total heat release curve, under different TCRs with SC of 72%, is shown in Figure 5f. It can be observed in the figure that the total amount of hydration heat release increases with the decrease in TCR. At 72 h, the difference reaches the maximum, and the whole heat release from TCR = 12 to TCR = 10, TCR = 8, TCR = 6, and TCR = 4 increases by 3.1 J/g, 3.3 J/g, 2.2 J/g, and 7.5 J/g, respectively. This shows that with the decrease in TCR, the water–cement ratio decreases correspondingly, which is conducive to the hydration reaction. As a result, the hydration reaction speed is faster, and more hydration products that can provide strength for CSTB can be produced simultaneously.

#### 3.2.2. Effect of SC on the Heat Exotherm of CSTB Hydration

As shown in Figure 6, SC’s hydration heat release rate increased from 68% to 72% under the same TCR and the total heat release amount of CSTB when TCR = 4. It can be observed in Figure 6a–e that the exothermic hydration curves of different SCs are similar under a given TCR—taking TCR = 4 as an example for analysis. The data in Table 4 show that SC = 68% enters the induction phase after 1.5 h and enters the acceleration phase after 4.9 h. Although the SC increased to 72%, in the hydration heat release curve, the time to join the acceleration phase is only 0.1 h ahead of schedule, and the duration of the acceleration phase is only 0.3 h ahead. In addition, it can be seen from the hydration heat release curve that the process of the acceleration stage is only 1.7 h. The appearance time of the second exothermic peak was delayed by 0.5 h. The third exothermic peak was delayed by 1.7 h. The changes in total heat release in different periods in Table 5 show that the complete heat release Q did not change significantly in the early stage. For example, when T = 18 h, Q only increased by about 13.0% as SC rose from 68% to 72%. The above experimental results show that, compared with the effect of TCR on CSTB, SC has a limited impact in the early stage. This is because when TCR is constant (cement content is consistent), the change in sulfate ion concentration, caused by the shift in SC, is small. Therefore, although SC increases from 68% to 72%, there was only a small difference in the time of sulfate depletion point. 

Meanwhile, when the cement content is constant, the increase in SC increases W/C to a certain extent. However, its impact is not as great as the influence of TCR on W/C. Therefore, the difference in hydration exothermic curve under different SC is not as significant as that under TCR. However, with time, the influence of SC will be more evident due to the different reaction rates in the slow reaction stage. For example, when t = 72 h, SC = 68% increases to 72%, and Q increases by 25.1%.

#### 3.2.3. Effect of TCR and SC Coupling on Hydration Heat Release of CSTB

The change in TCR and SC will cause the shift of W/C, which is proportional to TCR and inversely proportional to SC. W/C is an essential parameter for studying the cementation of cement-based materials. It can affect hydration rate, microstructure [67], and porosity [68]. Table 6 lists the changes in W/C by different TCR and SC, and Figure 7 further shows the changes in the total hydration heat of CSTB when W/C changes. It can be observed from Table 6 that the ratio of W/C increases by about 0.8 with each gradient of TCR, and decreases by only 0.2–0.3 with each rise in SC. It can be seen from Figure 7 that the total amount of heat released is inversely proportional to W/C. W/C changes faster with the increase in TCR, and the total amount of heat released decreases more obviously. W/C changes little with the rise in SC, and the change in entire heat release was insignificant. Therefore, the effect of TCR on the hydration process is more significant than that of SC.

Besides the effect of TCR on the hydration heat release of CSTB due to SC can be attributed to W/C, there may be other reasons. For example, the change in TCR will have an impact on cement and tailings in CSTB. The decrease in TCR reduces the content of tailings and leads to the reduction in nucleation sites of C-S-H/CH. However, compared with the increase in cement content, the decrease in nucleation sites may have a limited impact. The nucleation sites provided by tailings may have reached saturation when TCR = 4. Therefore, with the decline in TCR, there is no reduction in hydration heat release [66]. The increase in SC leads to the decrease in W/C and increases the content of cement and tailings per unit volume. The spacing between particles in CSTB is smaller, the dispersion effect on cement and tailings is weakened, and the hydration reaction speed increases. Therefore, the heat release rate and total heat release are increased [69,70].

Figure 8 displays the relationship between the total heat release of hydration in 72 h and the 3d compressive strength. The results show a positive correlation between the intensity and the total heat release and the relationship equations obtained by fitting, which is consistent with the conclusions of other scholars [47,64,71].

### 3.3. Hydration Products and Microstructure Analysis

The strength development of CSTB is closely related to the hydration products formed [72]. After CSTB is mixed with water, calcium silicate in cement is hydrated to create hydration products such as CH, C-S-H, and AFt. The XRD and SEM images of CSTB can analyze the microstructure of CSTB, thus revealing the visible strength change in CSTB from mineral composition and microscopic morphology.

#### 3.3.1. Effect of TCR on Hydration Products and Microstructure of CSTB

Figure 9 and Figure 10 show the XRD and SEM of CSTB samples at different curing ages when SC = 72%. It can be observed in the XRD pattern that the 3-day and 7-day ages mainly contain peaks corresponding to CH, C-S-H, and AFt, among which CH and C-S-H are due to the hydration reaction of calcium silicate, as shown in Equations (3) and (4):(3)$$3CaO\cdot SiO_{2}+H_{2}0\to xCaO\cdot SiO_{2}\cdot yH_{2}O(C-S-H)+Ca{\left(OH\right)}_{2}$$
(4)$$2CaO\cdot SiO_{2}+H_{2}0\to xCaO\cdot SiO_{2}\cdot yH_{2}O(C-S-H)+Ca{\left(OH\right)}_{2}$$

Since the generated CH provides an alkaline environment for CSTB, CH reacts with SiO_2_ in tailings in the alkaline environment, as shown in Equation (5). At the same time, with the participation of water, CH reacts with Al_2_O_3_ in tailings and CaSO_4_ in cement to form AFt through Equation (6) [73]:(5)$$xCa{\left(OH\right)}_{2}+SiO_{2}+\left(y-x\right)H_{2}O\to xCaO\cdot SiO_{2}\cdot yH_{2}O\left(C-S-H\right)$$
(6)$$Al_{2}O_{3}+3Ca{\left(OH\right)}_{2}+3CaSO_{4}\cdot 2H_{2}O+23H_{2}O\to 3CaO\cdot Al_{2}O_{3}\cdot 3CaSO_{4}\cdot 32H_{2}O\left(AFt\right)$$

The peaks corresponding to the hydration products can be observed in Figure 9, but the peak intensities are not wholly consistent. For example, at the diffraction angle of 34°2-theta, CH’s maximum relative peak intensity is TCR = 4, and the minimum peak intensity is TCR = 12. Even the peak cannot be observed. This indicates that more CH is generated at low-TCR conditions than high-TCR conditions. This phenomenon exists, not only for CH, but also for other hydration products.

The hydration products generated and the pores between tailings particles can be observed from the SEM of Figure 10 (a and c are TCR = 4; b and d are TCR = 12). In the case of TCR = 12, it can be seen that there are a lot of pores between tailings particles. When TCR = 4, AFt can be perceived on the surface of CSTB particles and connected. Hydration products wrap tailings particles, and pores are effectively filled. Only a few pores can be noted. At the same time, it can be observed that when the curing time reaches 7 days, the structure is dense.

The reason for the above phenomenon is that with the decrease in TCR, more cement can participate in the hydration reaction, and as such the hydration reaction speed is faster. In the same curing time, more hydration products are produced. That is, the relative intensity value of each hydration product in the XRD diffraction pattern is more significant, and more hydration products can be observed in the SEM image. After the hydration product is generated, it can effectively fill the pores between the tailings particles and play a role in connecting the tailings particles. Therefore, the pores observed at the lower TCR are relatively small, and the internal structure is more compact [74]. The compressive strength of CSTB is closely related to its internal pores and hydration products. Therefore, the compressive strength of CSTB with low TCR is higher than that of CSTB with high TCR. The strength changes and hydration heat analysis results mentioned above have been further confirmed.

#### 3.3.2. Effect of SC on Hydration Products and Microstructure of CSTB

As shown in Figure 11 and Figure 12 (a and c are SC = 68%; b and d are SC = 76%), it can be observed that the hydration products of CSTB increase with the increase in SC. For example, SC increases from 68% to 76%, and C-S-H peak intensity, corresponding to 31.7° 2-theta, gradually increases. AFt corresponding to 18.8 ° 2-theta and CH corresponding to 34° 2-theta also follow the same law. The higher the relative intensity of the peak, the more the hydration product is produced, and the hydration product increases with the increase in SC.

By observing SEM images, it can be observed that when SC is 68%, the formation of needlelike AFt is not seen when its age is 3 days, and only a tiny amount of needlelike AFt and flocculent gelled product C-S-H can be observed when the age increases to 7 days. When SC increased to 76%, the number of hydration products that could be observed in 3 days was higher than that in 7 days when SC was 68%, which was the same as the conclusion obtained by XRD.

The above phenomenon can be attributed to tailings, cement, and water reaction after mixing with the increase in SC. The water–cement ratio decreases, and the relative concentration of cement increases. The rise in concentration is conducive to producing more hydration products. The XRD diffraction pattern’s relative intensity values of C-S-H and CH are larger. These products will fill the pores of the particles in the backfill body, increase the compactness of the tailing’s backfill body and reduce the pore spacing, so the CSTB strength with higher SC will also be higher.

#### 3.3.3. Effect of Coupling Action of TCR and SC on Hydration Products and Microstructure of CSTB

TCR and SC do not affect the composition of hydration products of CSTB, and both will produce hydration products such as CH, C-S-H, and AFt that can provide strength for CSTB. The difference lies in the number of hydration products built. Hydration products increase with the decrease in TCR, and at the same time, they grow with the increase in SC. TCR and SC jointly determine the number of hydration products of CSTB.

At the same time, TCR and SC also impact the microstructure of CSTB, and the formation of hydration products can fill the internal pores of CSTB. TCR and SC can affect the appearance of hydration products and their structure. When SC is constant, the microstructure of CSTB with lower TCR is denser. When TCR is consistent, the microstructure of CSTB with higher SC is more compact.

## 4. Conclusions

In this paper, the influence of TCR and SC on the early mechanical strength, hydration characteristics, and microstructure of CSTB and the relationship between them are studied, and the following conclusions are reached:

The early intensity of CSTB increased with the decrease in TCR and the increase in SC. Still, the early power of CSTB had different sensitivity to TCR and SC and was more susceptible to TCR. The results show that the early strength of CSTB is exponentially related to TCR or SC, except that TCR = 4 at the curing age of 3 days.

With the increase in TCR, the peak values of the second and third hydration exothermic peaks gradually decreased and appeared earlier. The second exothermic peak disappeared when TCR = 12, the exothermic hydration rate slowly fell, the duration of the induction phase was prolonged, the acceleration phase was shortened, and the total heat release decreased.

The second and third hydration exothermic peaks increased with SC, and the occurrence time was delayed. The exothermic hydration rate gradually increased, but TCR did not significantly affect the difference. The total exothermic amount began to show a significant difference 18 h after the start of the test.

The main hydration products of CSTB are CH, C-S-H, and AFt. With the decrease in TCR, there are more hydration products, the corresponding XRD diffraction peak intensity is higher, the densification degree of the matrix in SEM image is enhanced, and the internal pores are reduced. The hydration products also increased with the increase in SC, which was consistent with the decrease in TCR.

The early strength, hydration heat, hydration products, and microstructure of CSTB are also affected by each other. The exothermic hydration rate of CSTB materials with low TCR and high SC is faster. The total amount of exothermic hydration increases, indicating that more hydration reactions are carried out and will generate more hydration products. These hydrates can better fill the gap between the CSTB particles, make the internal structure of CSTB more compact, and finally show higher CSTB strength.

## Figures and Tables

**Figure 1 materials-15-08846-f001:**
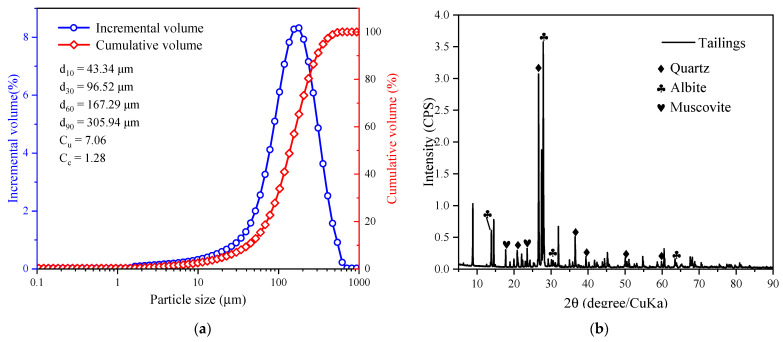
(**a**) Particle size distributions and (**b**) XRD patterns of tailings.

**Figure 2 materials-15-08846-f002:**
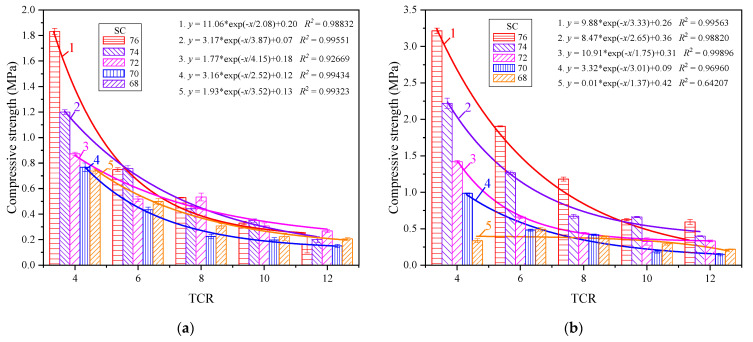
Compressive strength of CSTB at different TCRs with curing age of 3 d (**a**) and 7 d (**b**).

**Figure 3 materials-15-08846-f003:**
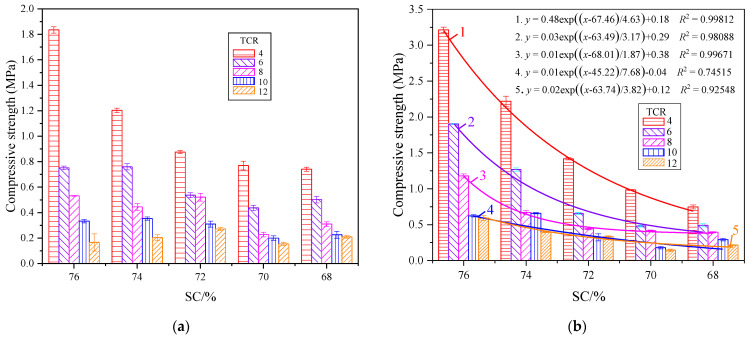
Compressive strength of CSTB at different SCs with curing age of 3 d (**a**) and 7 d (**b**).

**Figure 4 materials-15-08846-f004:**
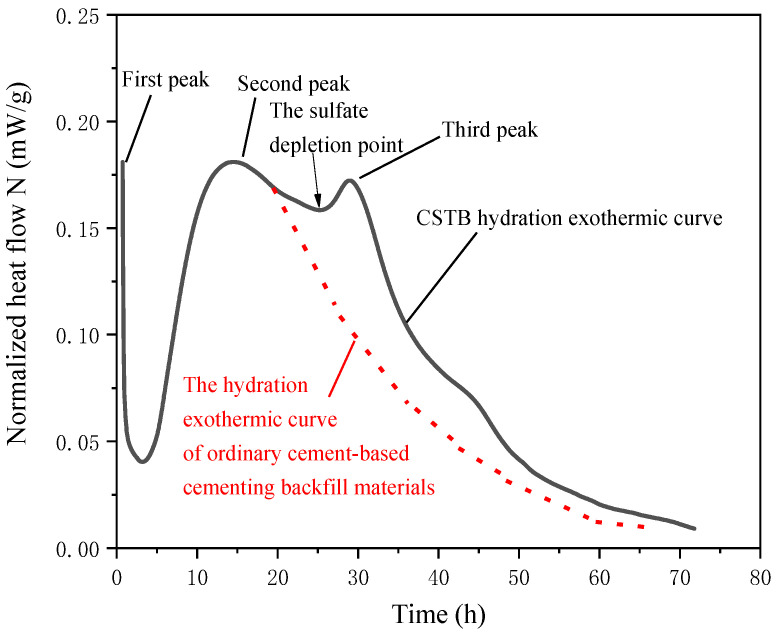
Differ exothermic hydration curves between ordinary cement-based materials [59] and CSTB.

**Figure 5 materials-15-08846-f005:**
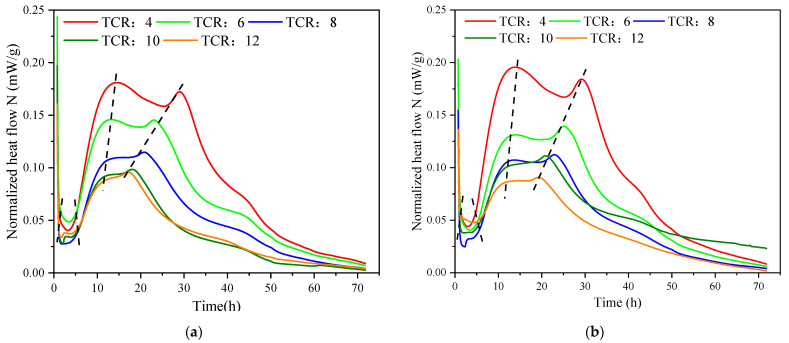
Hydration heat release rate curve: (**a**) SC = 68%, (**b**) SC = 70%, (**c**) SC = 72%, (**d**) SC = 74%, (**e**) SC = 76%, and total heat release: (**f**) SC = 72% of CSTB under different TCRs.

**Figure 6 materials-15-08846-f006:**
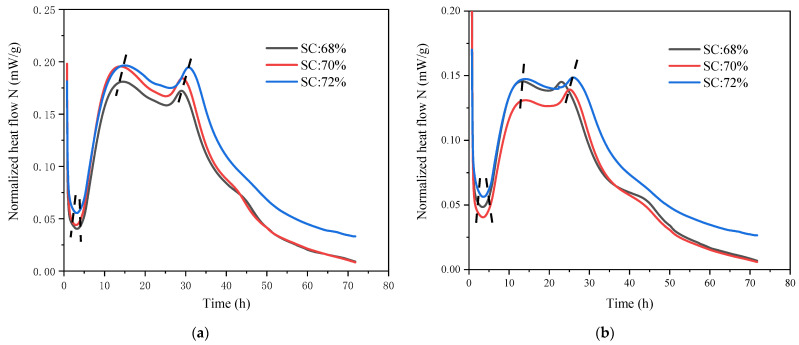
Hydration heat release rate curve: (**a**) TCR = 4, (**b**) TCR = 6, (**c**) TCR = 8, (**d**) TCR = 10, (**e**) TCR = 12 and total heat release: (**f**) TCR = 4 of CSTB under different SCs.

**Figure 7 materials-15-08846-f007:**
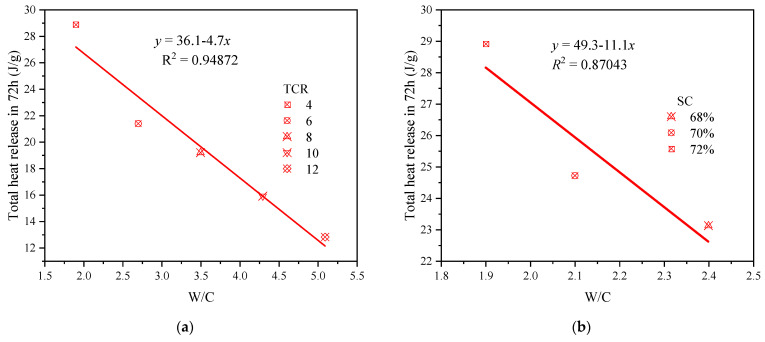
The effect of W/C on the total heat release Q under certain conditions: (**a**) SC = 72%; (**b**) TCR = 4.

**Figure 8 materials-15-08846-f008:**
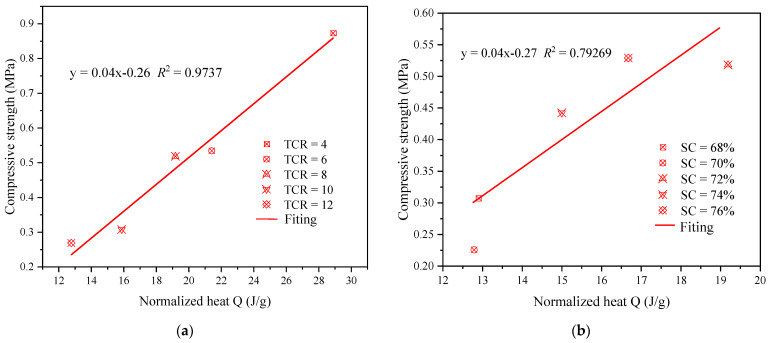
Hydration heat versus the compressive strength values at 3 days of CSTB: (**a**) SC = 72%; (**b**) TCR = 8.

**Figure 9 materials-15-08846-f009:**
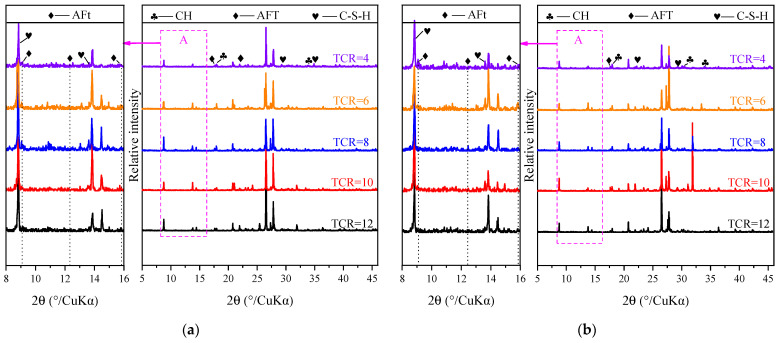
XRD patterns of CSTB in 3d (**a**) and 7d (**b**) under SC = 72%.

**Figure 10 materials-15-08846-f010:**
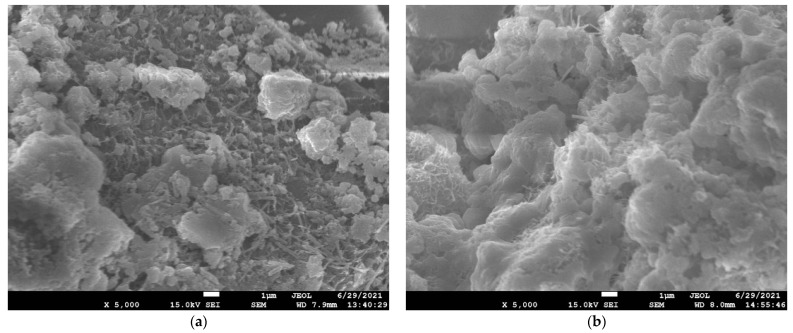
SEM patterns of CSTB in 3 d ((**a**) TCR = 4; (**b**) TCR = 12) and 7 d ((**c**) TCR = 4; (**d**) TCR = 12) under SC = 72%.

**Figure 11 materials-15-08846-f011:**
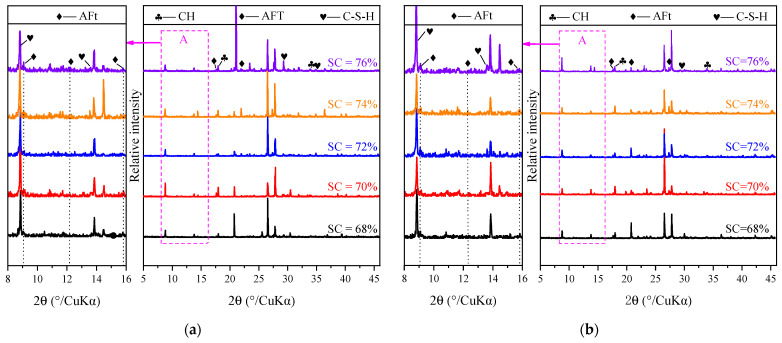
XRD patterns of CSTB in 3 d (**a**) and 7 d (**b**) under TCR = 4.

**Figure 12 materials-15-08846-f012:**
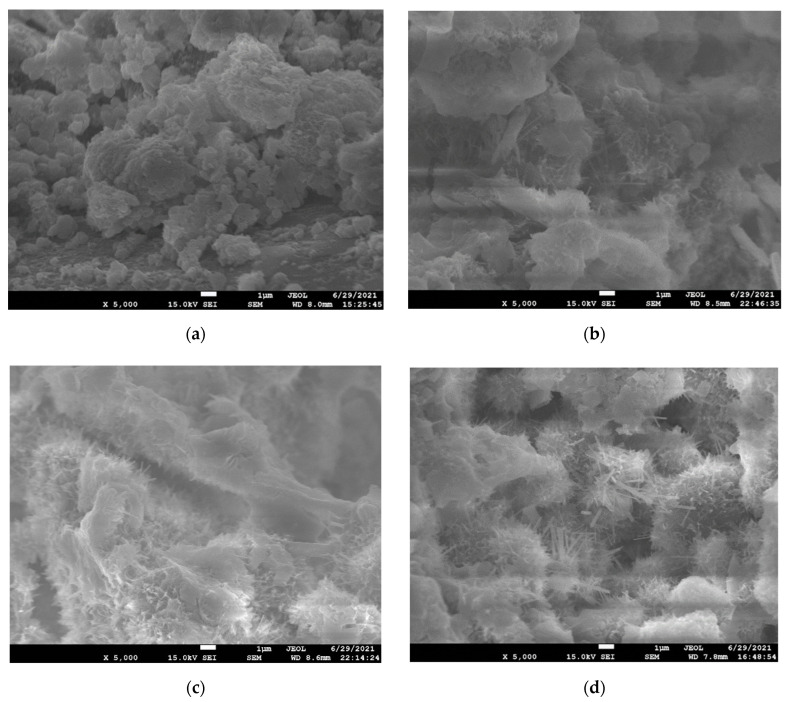
SEM patterns of CSTB in 3 d ((**a**) SC = 68%; (**b**) SC = 76%) and 7 d ((**c**) SC = 68%; (**d**) SC = 76%) under TCR = 4.

**Table 1 materials-15-08846-t001:** Chemical composition of cement and tailings.

Composition	CaO	SiO_2_	MgO	Fe_2_O_3_	Al_2_O_3_	SO_3_	TiO_2_	CuO	Na_2_O	Loss
Cement	51.84	27.75	4.88	2.94	7.65	1.06	0.24	0.02	0.18	4.76
Tailings	0.66	70.62	0.12	0.50	14.52	0.24	0.02	0.01	7.38	11.65

**Table 2 materials-15-08846-t002:** Experimental schemes.

SC/%	TCR	Material Quality/%		
		Tailing	Cement	Water
68	4:1	54.40	13.60	32
	6:1	58.29	9.71	32
	8:1	60.44	7.56	32
	10:1	61.82	6.18	32
	12:1	62.77	5.23	32
70	4:1	56.00	14.00	30
	6:1	60.00	10.00	30
	8:1	62.22	7.78	30
	10:1	63.64	6.36	30
	12:1	64.61	5.39	30
72	4:1	57.60	14.40	28
	6:1	61.71	10.29	28
	8:1	64.00	8.00	28
	10:1	65.45	6.55	28
	12:1	66.46	5.54	28
74	4:1	59.20	14.80	26
	6:1	63.43	10.57	26
	8:1	65.78	8.22	26
	10:1	67.27	6.73	26
	12:1	68.31	5.69	26
76	4:1	60.80	15.20	24
	6:1	65.14	10.86	24
	8:1	67.56	8.44	24
	10:1	69.09	6.91	24
	12:1	70.15	5.85	24

**Table 3 materials-15-08846-t003:** Variation in hydration exothermic rate for different TCRs (h).

TCRs	Induction Stage	Acceleration Stage	Second Hydration Peak	Third Hydration Peak	Deceleration Stage
4	1.4	5.1	15.1	30.8	33.6
6	1.6	5.8	14.1	26.1	28.9
8	1.3	5.8	13.7	23.6	27.7
10	1.2	6.5	14.0	21.0	26.5
12	1.1	6.6	None	20.3	24.2

**Table 4 materials-15-08846-t004:** Time of hydration exothermic phase of different SCs.

SCs	Induction Stage (h)	Acceleration Stage (h)	Second Hydration Peak (h)	Third Hydration Peak (h)	Deceleration Stage (h)
68%	1.5	4.9	31.7	3.4	26.8
70%	1.5	4.7	31.9	3.2	27.2
72%	1.4	5.1	33.6	3.7	28.5

**Table 5 materials-15-08846-t005:** Total heat release Q (J/g) under different SCs.

SCs	18 h	36 h	54 h	72 h
68%	7.7	17.8	21.9	23.1
70%	8.6	19.3	23.5	24.7
72%	8.7	20.4	26.2	28.9

**Table 6 materials-15-08846-t006:** Variation in W/C with TCR and SC under given conditions.

TCRs/SCs	TCR = 4	TCR = 6	TCR = 8	TCR = 10	TCR = 12	SC = 68%	SC = 70%	SC = 72%
W/C	1.9	2.7	3.5	4.3	5.1	2.4	2.1	1.9

## Data Availability

The data used to support the findings of this study are included in the article.

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
