# Peer review of "Experimental Study on Early Strength and Hydration Heat of Spodumene Tailings Cemented Backfill Materials"

_materials, 2022, doi:10.3390/ma15248846_

Round 1

Reviewer 1 Report

1)      “Accordingly, this paper proposes to prepare spodumene tailings cemented backfill (CSTB) by using cement and spodumene tailings, uniaxial compressive strength, heat of hydration, XRD and SEM tests were used to study the mechanical behavior, hydration exotherm, hydration products and microstructure morphology of early CSTB with different tailing-cement ratio (TCR) and solid mass concentration (SC) to promote the resource utilization of spodumene tailings.”

Please review the sentence.

2)      “Lack of knowledge about the fundamental properties of spodumene tail-ings backfill (CSTB).”

What are the conclusions about existing studies? And what is the differential of this research concerning existing studies? This must be made clear in the manuscript.

3)      Figure 1: Please indicate the files (e.g., ICDD) and database used to identify the peaks in the diffractograms.

4)      “For the purpose of the experiment, the tailings and cement ratios (TCR) were prepared as 4:1, 6:1, 8:1, 10:1, 12:1 and the solid mass concentrations (SC) were 68%, 70%, 72%, 74%, 76% of the samples”.

How were these values ​​established? Based on the literature? This information must be provided.

5)      Better describe the isothermal calorimetry test. Insert information such as sample mass, and analysis time. Were the values ​​normalized about the cement mass? These informations must be incorporated into the manuscript.

6)      “After grinding until there is no particle feel-ing, use Bruker AXS D8 X-ray diffractometer produced by Bruker company in Germany to determine the type of hydration products of CSTB by XRD.”

Describe the test conditions (step size, scan interval, etc.). Was the sample sieved?

7)      Insert a table with the evaluated compositions detailing the cement, tailings, and water amount.

8)      Figure 3: Add the standard deviation of the compressive strength results.

9)      Discussion of compressive strength results: Justify the results, correlate with the other tests (e.g., calorimetry), and compare with literature.

10)   Figure 4: Add the standard deviation of the compressive strength results.

11)   “One part is the hydration reaction of tricalcium silicate (C3S) contained in cement, that is, the second exothermic peak of ordi-nary cement-based materials, and the other part is the hydration reaction of tailings, that is, the third exothermic peak, which is similar to the hydration exothermic curve produced by adding blast furnace slag [49] or iron tailings [50] to cement; Second, the emergence of the new exothermic peak is caused by the transformation of ettringite (AFt) into mono-sulfur calcium sulfoaluminate (AFm)[51,52], and the concentration of sulfate plays an im-portant role in this process”.

Please, see Fig. 7 in the following article: https://doi.org/10.1016/j.conbuildmat.2021.122428 .

Check if the statement is correct. I believe the third peak is associated with the renewed C3A hydration. The conversion of ettringite into AFm phases is usually identified in the "shoulder" of the heat flow, which in the case of the study occurs between 40-50 hours.

12)   Identify the sulfate depletion point in Fig. 5

(see https://doi.org/10.1016/j.conbuildmat.2021.122428 .)

13)   Insert a zoom of diffractograms between 9 - 16 2θ to better visualize the AFm phases.

14)   Check the peaks assigned to C-S-H. the C-S-H is an amorphous or semi-crystalline phase.

15)   “From the SEM of Fig 10(a and c are TCR=4, b and d are TCR=12), the hydration products generated and the pores between tailings particles can be clearly observed. In the case of TCR=12, AFt content is less, and there are a lot of pores between tailings particles. However, when TCR=4, a large number of AFt spread over the surface of CSTB particles and connected with each other, tailing particles were wrapped by hydration products, and pores were effectively filled, so only a small number of pores could be observed. At the same time, it can be observed that when the curing reaches 7 days, the hydration products generated are increased, and a large amount of AFt is also generated in the pores.”

Please rewrite the sentence. Only one SEM image was recorded. Based on this image alone, it is impossible to say whether there are more or fewer AFm phases. For quantitative analysis from SEM, a large number of images is required.

Reviewer 2 Report

1. Explain the practical application of the work presented in the manuscript.

2. What is the reason of selecting the specific combination and composition of the materials selected in the work.

3. What other methods/ techniques could be adopted for the purpose explained in the manuscript and how the presented methodology is advantageous to those other methods.

4. Explain the future scope of the work.

Reviewer 3 Report

Please have a look at the attached file

Reviewer 4 Report

the review comments of the study is presented as follows.    

While TCR is abbreviated as "tailings-cement ratio" in the abstract, it is abbreviated as "sand to cement ratio" in the last paragraph of the introduction. Please fix this.

      Please explain in more detail the reason for the use of tailings within the scope of this study by interpreting the chemical contents and physical properties of the wastes presented in Table 1 and Figure 1.

      For a more clarified presentation, please add the mixture proportion table. 

      The titles of sections 3.1.1 and 3.1.2 are the same. Please check and correct.

In the introduction section of the study, it has been said that understanding the evolution law of microstructure correctly helps to analyse the law of early strength development so that the filler material controls the structure formation process and provides reasonable process parameters. But while reading the article, I cannot clearly see the  innovations brought by the study. What are the innovations that will contribute to the literature with early strength, exothermic hydration and microstructure analyses? What is the novelty of the study? Please explain clearly.

Round 2

Reviewer 1 Report

Accept in present form

Author Response

Dear Reviewer :

We submitted the revised version of the paper on December 1, but it was not submitted successfully due to improper operation. We apologize for the inconvenience, and now we have resubmitted.